# Characterization and In Vitro Efficacy against *Listeria monocytogenes* of a Newly Isolated Bacteriophage, ΦIZSAM-1

**DOI:** 10.3390/microorganisms9040731

**Published:** 2021-03-31

**Authors:** Silvia Scattolini, Daniela D’Angelantonio, Arianna Boni, Iolanda Mangone, Maurilia Marcacci, Noemi Battistelli, Krizia D’Agostino, Francesco Pomilio, Cesare Camma, Giacomo Migliorati, Giuseppe Aprea

**Affiliations:** 1Istituto Zooprofilattico Sperimentale dell’Abruzzo e del Molise “G. Caporale”, 64100 Teramo, Italy; s.scattolini@izs.it (S.S.); d.dangelantonio@izs.it (D.D.); i.mangone@izs.it (I.M.); m.marcacci@izs.it (M.M.); k.dagostino@izs.it (K.D.); f.pomilio@izs.it (F.P.); c.camma@izs.it (C.C.); g.migliorati@izs.it (G.M.); 2Istituto Superiore di Sanità, 00161 Rome, Italy; arianna.boni@iss.it; 3Faculty of Bioscience and Technology for Food, Agriculture and Environment, University of Teramo, 64100 Teramo, Italy; nbattistelli@unite.it

**Keywords:** bacteriophage, foodborne pathogen, *Listeria monocytogenes*, in vitro efficacy test

## Abstract

*Listeria monocytogenes* is a bacterial pathogen responsible of listeriosis, a disease that in humans is often related to the contamination of ready-to-eat foods. Phages are candidate biodecontaminants of pathogenic bacteria thanks to their ability to lyse prokaryotes while being safe for eukaryotic cells. In this study, ΦIZSAM-1 was isolated from the drain-waters of an Italian blue cheese plant and showed lytic activity against antimicrobial resistant *Listeria monocytogenes* strains. This phage was subjected to purification and in vitro efficacy tests. The results showed that at multiplicities of infection (MOIs) ≤ 1, phages were able to keep *Listeria monocytogenes* at low optical density values up to 8 h, with bacterial counts ranging from 1.02 to 3.96 log_10_ units lower than the control. Besides, ΦIZSAM-1 was further characterized, showing 25 principal proteins (sodium dodecyl sulfate polyacrylamide gel electrophoresis profile) and a genome of approximately 50 kilo base pairs. Moreover, this study describes a new approach to phage isolation for applications in *Listeria*
*monocytogenes* biocontrol in food production. In particular, the authors believe that the selection of phages from the same environments where pathogens live could represent a new approach to successfully integrating the control measures in an innovative, cost effective, safe and environmentally friendly way.

## 1. Introduction

*Listeria monocytogenes* (*L. monocytogenes*) is a Gram-positive foodborne pathogen that causes listeriosis. The disease is a serious problem in Europe and was the fifth most common zoonosis for number of reported confirmed human cases (*n* = 2621) in 2019, with a fatality rate of 17.6% [1]. Bacteriophages are the simplest lifeforms on earth and their number on our planet is estimated to be 10^31^ [2]. They are the natural enemies of bacteria and play a key role in the biocontrol of prokaryotic cells in order to avoid their excessive multiplication in the environment. For this reason, together with their selective activity against specific hosts/strains and due to being safe towards eukaryotic cells, their role as potential bio-decontaminants against pathogenic foodborne bacteria has recently become more frequently investigated by the scientific community [3]. Moreover, phage-based biocontrol strategies have received increasing interest in recent years also due to the relatively low cost of their preparations and applications [4].

*L. monocytogenes* can be efficiently controlled by the use of virulent bacteriophages in cheese [5,6,7,8,9]. Some countries like the USA, Canada, and Switzerland allow the use of phages in food production and a few phage-based formulations against *Listeria* are available in commerce (e.g., ListShield, a mix of six different bacteriophages and its activity has been tested specifically on fruits; Listex™ P100, which is constituted by the bacteriophage P100) [4]. Europe, which, in contrast, has tough regulations considering phage use safe, is still quite skeptical about their applications along the food chains. In particular, among the limitations of the use of bacteriophages as bio-decontaminants in food productions, experts from the European Food Safety Authority (EFSA) highlighted the necessity of more specific studies that demonstrate the phage efficacy above all against strains that originate from the same food matrixes that have to be phage-treated [10]. In this research, the authors monitored the persistence of *L. monocytogenes* in an Italian blue cheese plant during four surveys and the pathogen was constantly found in many production areas (*L. monocytogenes* serovar 1/2a, 1/2b and 4b). During the fourth survey, the drain-waters were also assessed for phage isolation against *L. monocytogenes*. In particular, one phage (ΦIZSAM-1) was isolated from the floor-drain in the cheese salting area and morphologically characterized (family *Siphoviridae*, order *Caudovirales* [11] (Figure 1).

In this study, ΦIZSAM-1 has been further characterized (genome and protein sizes) and showed a wide host range activity against 21 strains of *L. monocytogenes* characterized by different serovars and food origin, comprising the strains isolated from the blue cheese plant under investigation. Among the 21 strains sensitive to ΦIZSAM-1, 12 were previously tested by the authors and they were resistant to antibiotics, some of which are commonly used in clinical settings, e.g., chloramphenicol, lincomycin, linedolid and nitrofurantoin [12]. Nowadays the increasing problem of multidrug resistance for the antibiotics largely used in animals and in humans is considered a serious threat in public health, in relation to the difficulties encountered in treating some bacterial infections with a consequential increase in the number of human deaths. The most recent European Union summary report (2020) on antimicrobial resistance (AMR) in zoonotic and indicator bacteria from humans, animals and food, among the antibiotics mostly used, highlighted the worrying levels of resistance to chloramphenicol, in particular [13]. This resistance is noted to be frequently developed by many pathogenic bacteria and it has been observed in all the animal species commonly bred for the production of foods for human consumption. In this scenario, the potential of using ΦIZSAM-1 against AMR *L. monocytogenes* strains in food production could be of great value and needs to be more deeply investigated. The phage efficacy was assessed in vitro against *L. monocytogenes* ATCC7644, by screening different ranges of multiplicities of infection (MOIs): 0.1, 1 and 10.

The aim of this work was to give a new approach to the fight against bacterial pathogens in food production based on the isolation and selection of naturally occurring phages from specific food plants, with the purpose of applying them in those environments, once these phages have been characterized and their efficacy successfully tested. The use of phages directly isolated from the food establishments could represent a valid support to successfully control bacterial contaminations, especially those derived from antimicrobial resistant (AMR) strains, in an innovative, cost effective, safe and environmentally friendly new way in order to reduce the extensive use of chemical agents and their consequent negative implications [14].

## 2. Materials and Methods

### 2.1. Samples

An Italian blue cheese plant was monitored 4 times over a period of 1 year for *L. monocytogenes* contamination. During the last survey, 4 drain-water samples were collected (Falcon, 50 mL) from different places in the plant (cheese making area n. 1 sample, salting area n. 2 samples and seasoning area n. 1 sample) and analyzed for phage isolation. Water samples were centrifuged at 2500× *g* gravity (*g*) for 10 min, filtered (0.45 µm) and stored at +4 °C until phage isolation assays.

### 2.2. L. Monocytogenes Strains (Hosts)

For phage isolation and host range assessment, 21 different *L. monocytogenes* strains (Table 1) were singularly cultured (0.1 mL) in 10 mL of Brain Heart Infusion broth (BHI broth) (Oxoid^TM^, Hampshire, UK) and incubated at 30 °C for 16 h in order to reach cell concentrations of about 10^9^ colony forming units (cfu)/mL. Among the strains, *L. monocytogenes* ATCC7644 was used for phage propagation and for in vitro efficacy tests.

### 2.3. Phage Isolation, Plaque Purification and Host Range Assessment

Phage isolation and host range assessment were carried out by using double-layer agar techniques according to Spears et al. (2008) [15], with some modifications. Briefly, 100 microliters (µL) of each host were individually added to 4 mL of BHI soft agar (BHI broth + 7 g/L of agar—Oxoid^TM^, Hampshire, UK) supplemented with 100 µL of 1M CaCl_2_ and 1 M MgSO_4_. The soft agar was gently vortexed and poured into a BHI agar (BHI broth + 15 g/L agar—Oxoid^TM^, Hampshire, UK) plate. The presence of *Listeria* phages was assessed by spotting 10 µL of samples over the plate and incubating overnight at 30 °C. The zones of clearance from the plates were cut carefully from the agar using a sterile loop, added to 3 mL of SM broth (0.05 M TRIS, 0.1 M NaCl, 0.008 M MgSO_4_, 0.01% (weight in volume) gelatin, pH 7.5) and left for 89 h at ambient temperature. Then, the phage suspension was filtered (0.45 µm) and propagated to increase the phage titer up to 10^9^–10^10^ plaque forming units (pfu)/mL, according to Loessner and Busse (1990) [16]. For assuring the purification of the phage, this step was carried out 3 times consequently (3 step plaque purification assay) [17]. Only the purified ΦIZSAM-1 was further assessed for host range, genome/protein analysis and in vitro efficacy test.

### 2.4. In Vitro Efficacy Test

The test was performed in 24-well plates. *L. monocytogenes* suspensions were serially diluted to achieve a final concentration of 10^6^ cfu/mL, and then 1.8 mL were distributed in each well [18]. Two hundred µL of ΦIZSAM-1 was added in each well to achieve a final concentration ranging from 10^7^ to 10^5^ pfu/mL (MOIs 10, 1 and 0.1). In the *Listeria* control well, 0.2 mL of BHI broth without phages was added. The negative control consisted of 2 mL of BHI broth placed in each plate under assessment. At time 0 (T0h), some aliquots were taken from the wells in order to confirm *Listeria* and phage titer. The plates were incubated at 30 °C and the optical density (OD) 600 nm (Eppendorf, Hamburg, Germany) was read at T2h, T4h, T6h, T8h, T21h30min, T24h, T26h30min, T29h30min and T33h. The experiments were carried out in triplicate.

### 2.5. Phage Concentration and DNA Extraction

For more in depth genome and protein characterization, ΦIZSAM-1 was subjected to concentration as described by Boulanger (2009) [19] with substantial modifications. In brief, after 2 cycles of propagation to increase phage titer, the phage suspension was subjected to 2 cycles of freeze–thawing, centrifugation (18,100× *g* for 30 min (Sorvall RC-5C plus and SLC-3000 rotor, Hanaou, Germany), filtration (0.45 µm), chloroform treatment (1%) and further centrifugation (3000× *g* for 10 min). Polyethylene glycol (PEG) 8000 (Merck, Darmstadt, Germany) was then added to the suspension (10%) and stored overnight at +4 °C. The PEG solution was then centrifuged at 21,100× *g* for 25 min and the pellets were reconstituted in SM broth and stored at +4 °C overnight. After addition of chloroform (1%), the suspension was centrifuged at 3000× *g* for 10 min and filtered (0.45 µm). For genome analysis, part of the phage suspension was subjected to DnaseI (New England Biolabs, Bishops Stortford, UK) treatment according to the manufacturer’s instructions and the DNA was extracted with High-Pure DNA isolation kit (Roche, Hamburg, Germany). Instead, for the sodium dodecyl sulfate polyacrylamide gel electrophoresis (SDS-PAGE), the phage suspension was subjected to 4 cycles of freeze–thawing, centrifuged at 32,000× *g* (Beckman XL 70 and SW40Ti rotor, Krefeld, Germany) for 30 min and the pellet was suspended with 20 µL of SM broth (0.05 M TRIS, 0.1 M NaCl, 0.008 M MgSO4, 0.01% (weight in volume) gelatin, pH 7.5).

### 2.6. Phage Genome Sequencing and Sequence Analysis

Extracted DNA was quantified by using the Qubit^R^ DNA HS Assay Kit (Thermo Fisher Scientific, Waltham, MA, USA). One nanogram was used for library preparation and analyzed with Nextera XT Library Prep kit (Illumina Inc., San Diego, CA, USA) according to the manufacturer’s protocol. Sequencing was performed on Illumina NextSeq 500 platform using the NextSeq 500/550 Mid Output Reagent Cartridge v. 2 with 300 cycles and standard 150 base pairs (bp) paired-end reads. Reads were trimmed by an in-house script for adaptors removing and quality check, then they were de novo assembled using SPADES [20]. Obtained contigs were aligned against *nr* database [21] by BlastX [22] and annotated by PROKKA [23] to verify assembly consistence and to exclude *Listeria* contaminations or presence of chimeric results. Passing contigs were aligned to *nt* database by megaBlast [22] for similarity searching. The best matches were further aligned to assembled contigs by EMBOSS Stretcher tool [24].

### 2.7. Analysis of Phage Structural Proteins (SDS-PAGE)

The electrophoretic separation of proteins was performed with constant voltage at 200 V using current generator Power Ease 500 (Invitrogen, Carlsbad, CA, USA), polyacrylamide gel NuPAGE 4–12% Bis-Tris gel (Invitrogen, Carlsbad, CA, USA) and MES running buffer (Invitrogen, Carlsbad, CA, USA). Novex Sharp Prestained Protein Standard (Invitrogen, Carlsbad, CA, USA) was used as molecular weight marker. Samples were filled using NuPAGER LDS Sample Buffer (4×) and NuPAGER Sample Reducing Agent (10×) (Invitrogen, Carlsbad, CA, USA). After running, the gel was washed 3 times in deionized water to remove the buffer salts. Then, SimplyBlue™ SafeStain LC6065 (Pierce, Rockford, IL, USA) was added and the excess of staining was removed with deionized water. The gel was analyzed with ChemiDoc (Biorad, Hercules, CA, USA) using Image Lab™ Software (Biorad, Hercules, CA, USA).

## 3. Results and Discussions

One phage (ΦIZSAM-1) was isolated from the floor-drain in the salting area of an Italian blue cheese dairy factory. It showed a wide host range activity against 21 different strains of *L. monocytogenes*, chosen from international strain collections and from the biobank within the Italian National Reference Laboratory for *L. monocytogenes*. Moreover, it also showed lytic activity against 3 persistent *L. monocytogenes* strains (serovars 1/2a, 1/2b and 4b) isolated from the same plant where the phage was detected in the drains (Table 1). In a previous publication, the authors already mentioned an important finding related to the first 12 strains of *L. monocytogenes* in Table 1. Those strains, in fact, apart from being sensitive to ΦIZSAM-1, were also resistant to some antibiotics commonly used in therapy, e.g., chloramphenicol, lincomycin, linedolid and nitrofurantoin [12]. In this regard, an interesting finding was reported by EFSA experts in 2016 on the safety and efficacy of Listex™ P100 to be used against *L. monocytogenes* contaminations [10]. In particular, in this document, *L. monocytogenes* strains originally resistant to ciprofloxacin and erythromycin, reverted to being antibiotic-sensitive after infection with ΦP100. We also demonstrated a similar finding when putting under new-isolated-field phage pressure the AMR *Campylobacter jejuni* NCTC 12,662 strain [25]. The authors believe that this topic is of high importance, thus shedding light on potential useful applications of phages in therapy, when dealing with AMR pathogenic bacteria [12].

In Figure 2, the authors reported the results achieved from the in vitro assay at different MOIs of 0.1, 1 and 10, compared with the *Listeria* control.

In particular, ∆ OD values (OD *Listeria* control–OD phage-treated suspensions) at T21h30min were of 0.461 (MOI 10), 0.516 (MOI 1) and 0.600 (MOI 0.1). At MOI 10, 1 and 0.1 ΦIZSAM-1 showed an OD trend from T0 to T33h similar to the control, but with much lower OD values. In particular, ΦIZSAM-1 expressed the best results in terms of lower OD values at MOI 0.1, followed by MOI 1 and finally by MOI 10. *L. monocytogenes* (control) showed a growth peak at T21h30min (OD600nm 0.800), corresponding to a titer of 2.1 × 10^9^ cfu/mL. At the same time point, instead, the *Listeria* count was kept at 1.02 log_10_ units lower, compared to control, when treated with the phage at MOI 0.1 (OD600nm 0.200) (data not shown). In relation to phage replication, from T0 to T33h, ΦIZSAM-1 increased the titer of 4.54 log_10_ units at MOI 0.1 (data not shown). The ability of ΦIZSAM-1 to increase its titer at MOI ≤ 1 and the results obtained in relation to OD values associated with different MOIs suggest that the phage suspensions should be applied at final concentrations equal to or lower than the concentrations expected for the host in order to express the best anti-*Listeria* activity. In our laboratory experience, the “active mode” of phage application, as discussed by Cairns et al., 2009 [26], was verified to be more successful than the “passive mode”, and this is expressed by using lower MOIs (≤1). More in general, our findings are in accordance with Rong et al. (2014) [27] who demonstrated that the use of lower MOIs to reduce the presence of *Vibrio*
*parahaemolyticus* during the depuration of oysters resulted as the best choice. In contrast with this study, others showed that higher MOIs (>1) led to a higher reduction of pathogen loads, also depending on the type of food, with greater reduction of *Listeria* in liquid food compared to the solid ones [5,6,18,28,29]. Ayaz et al. (2018) [30] instead, showed that the application of different MOIs did not lead to significant differences in the bacterial counts. Our preliminary data need to be further assessed with trials focusing specifically on the efficacy of ΦIZSAM-1 applications on contaminated foods prior to deciding which MOI could be the best to apply in the future for food decontamination purposes. Furthermore, in our study, more than 40 of the *Listeria* colonies isolated from BHI broths at the end of the in vitro assay (33 h post phage-treatment) were tested for sensitivity to ΦIZSAM-1. These strains were still lysed after phage infection (data not shown). The authors are aware that 33 h phage-exposure is a short time to justify any phage resistance development; however, these preliminary results are in accordance with Carlton et al. (2005) [5] who never experienced the development of phage resistant *Listeria* strains isolated from cheeses when treating with low concentrations of ΦP100. Guenther et al. (2009) [6] also did not find any phage resistance development in *L. monocytogenes* strains treated with phages A511 and P100 in plating assays. In addition, similar results are reported in more recent papers on *L. monocytogenes* strains treated with phages [31,32,33]. Genome sequencing from ΦIZSAM-1 DNA returned 5 337 447 paired reads suitable for assembling. After checking for potential *Listeria* contamination and assembly consistence, a single contig of 50,021 nucleotides (phage genome approximately 50 kilos base pairs (kbp)) was obtained with a mean coverage higher than 4000×. Similarity searching returned *Listeria* ΦLP-101 (KJ094023) as the closest match, which is a phage with genome lengths of approximately 4377 kbp. The two genomes were linear and pairwise alignment showed a global identity of about 46% (coverage 54%). The annotation displayed 83 coding sequences with a length ranging from 110 to 383 nucleotides, covering more than the 92% of the genome with a gene density of 1.66 genes per kbp. Data about whole genome sequences and annotation from ΦIZSAM-1 cannot be submitted at the present because this material is undergoing a patent registration process. In relation to the protein profile, the SDS-PAGE of ΦIZSAM-1 revealed 5 major proteins with molecular weights of 150, 100, 40, 38 and 36 kDa and another 20 minor proteins (260, 145, 80, 75, 69, 65, 59, 58, 48, 32.5, 30, 28, 22.5, 20, 19, 18, 15, 14, 12.5, 11 kDa) (Figure 3).

## 4. Conclusions

This study presents a unique approach that aims at isolating and further characterize bacteriophages from food premises with the purpose of employing them in the same places against foodborne bacterial contaminations. This could help to overcome the general skepticism related to EFSA dissertations such as the opinion expressed in 2012, Tor 2 [10]. Moreover, for practical use, different MOIs have to be tested in vitro and on foods, since it is not always true that higher phage concentrations exhibit the best results in terms of efficacy. The potential of our phage to contrast AMR *Listeria* proliferations in vitro and the ability to avoid phage resistant mutants development deserve in-depth scientific insights in order to evaluate the possible use of ΦIZSAM-1 for *L. monocytogenes* biocontrol within cheese industrial facilities, as a tool to integrate/enforce good manufacturing practices in situ.

## Figures and Tables

**Figure 1 microorganisms-09-00731-f001:**
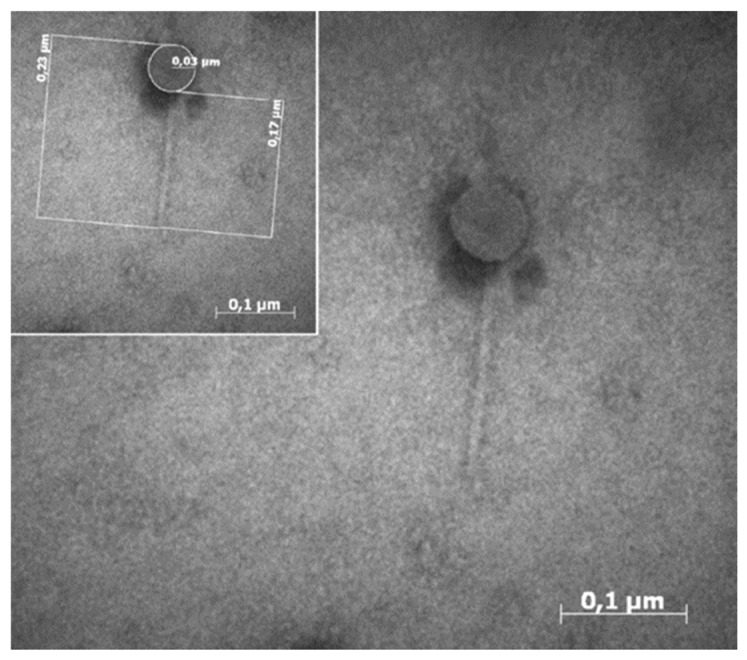
ΦIZSAM-1 under transmission electron microscope observation (50,000×) (integration from the source Aprea et al., 2015) [11].

**Figure 2 microorganisms-09-00731-f002:**
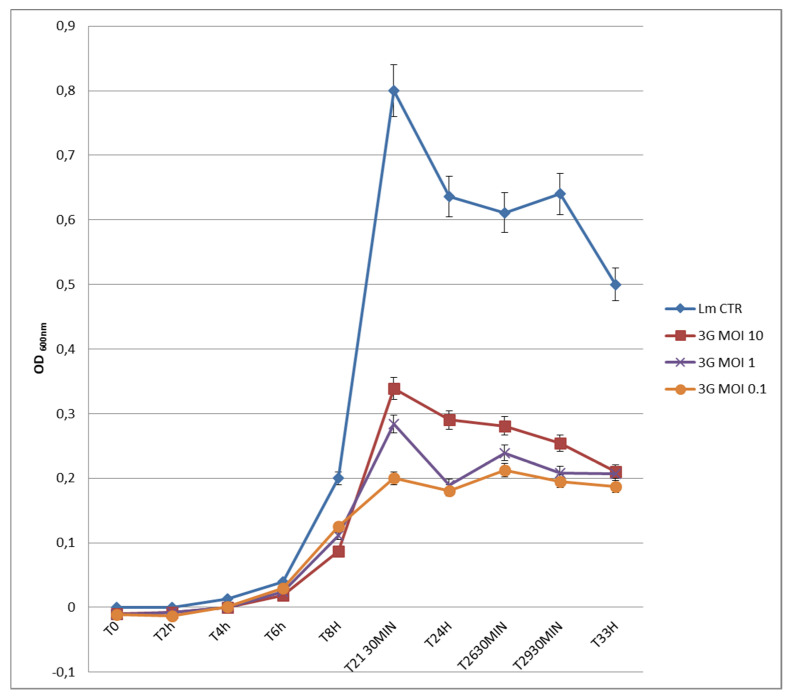
Optical density values of *L. monocytogenes* ATCC7644 challenged with ΦIZSAM-1 (3G) at different multiplicity of infections (MOIs) from T0 to T33h. Lm CTR: *L. monocytogenes* ATCC7644 untreated control; 3G MOI 10: *L. monocytogenes* ATCC7644 challenged with ΦIZSAM-1 MOI 10; 3G MOI 1: *L. monocytogenes* ATCC7644 challenged with ΦIZSAM-1 MOI 1; 3G MOI 0.1: *L. monocytogenes* ATCC7644 challenged with ΦIZSAM-1 MOI 0.1.

**Figure 3 microorganisms-09-00731-f003:**
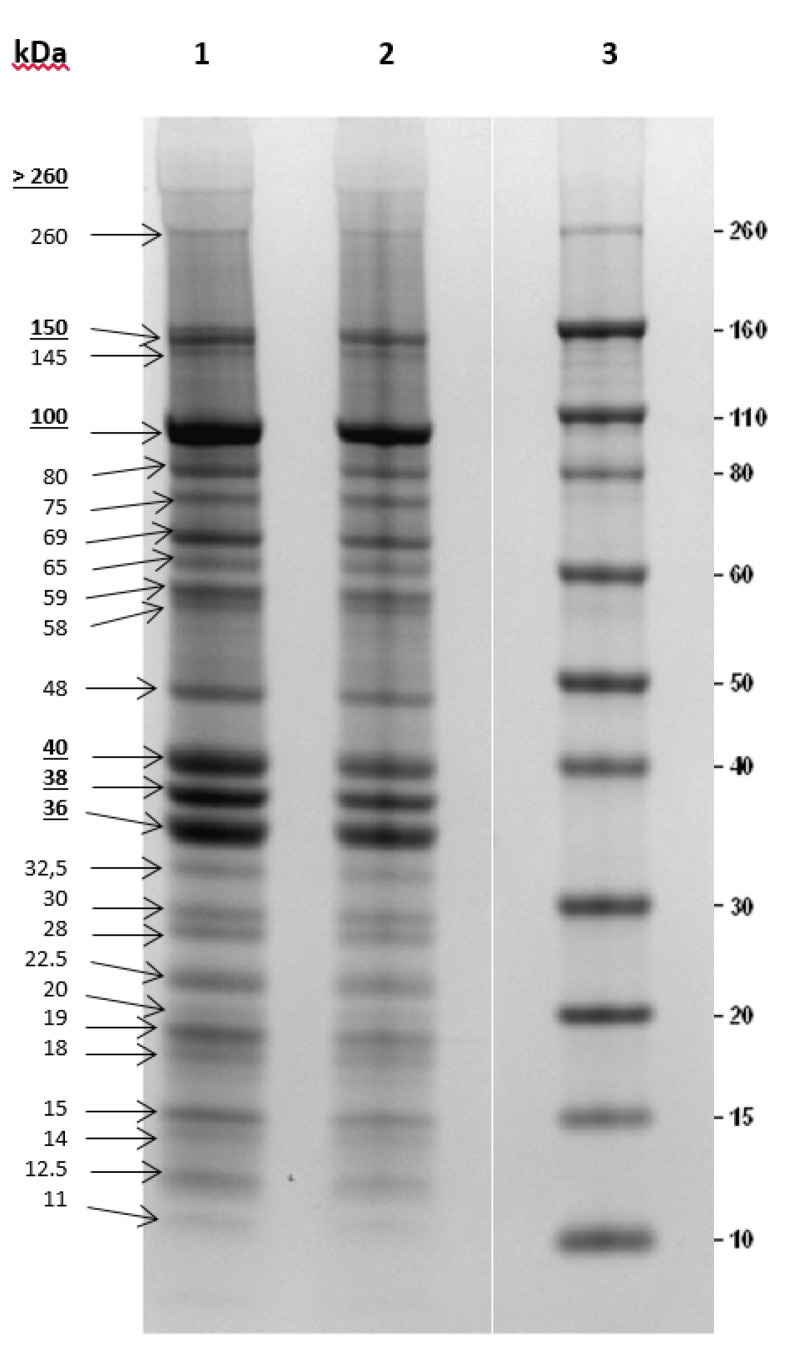
ΦIZSAM-1 structural proteins. Lane 1 and 2: ΦIZSAM-1; lane 3: Molecular weight protein marker.

**Table 1 microorganisms-09-00731-t001:** *Listeria monocytogenes* strains utilized as hosts for ΦIZSAM-1 isolation and host range analysis (updated table from Aprea et al., 2018) [12].

N.	Strain	Serovar	Origin
1	*L. monocytogenes* ATCC7644	1/2 c	ATCC
2	*L. monocytogenes*	1/2 b	Fresh pork sausage
3	*L. monocytogenes*	1/2 a	Chicken meat
4	*L. monocytogenes*	1/2 b	Bovine meat
5	*L. monocytogenes*	4 b	Fresh pork sausage
6	*L. monocytogenes*	1/2 c	Pork minced meat
7	*L. monocytogenes*	4 b	Pangasius fillet
8	*L. monocytogenes*	1/2 c	Bovine meat
9	*L. monocytogenes*	1/2 a	Bovine meat
10	*L. monocytogenes*	4 b	Smoked salmon
11	*L. monocytogenes*	4 b	Smoked salmon
12	*L. monocytogenes*	4 b	Human cepahlorachidian fluid
13	*L. monocytogenes* NCTC10887	1/2 b	NCTC
14	*L. monocytogenes* NCTC4883	4 c	NCTC
15	*L. monocytogenes* ATCC19115	4 b	ATCC
16	*L. monocytogenes* NCTC4886	1/2 a	NCTC
17	*L. monocytogenes* ATCC19114	4 a	ATCC
18	*L. monocytogenes*	2 a	Smoked salmon
19	*L. monocytogenes*	1/2 a	Blue cheese enstablishment (this research)
20	*L. monocytogenes*	1/2 b	Blue cheese enstablishment (this research)
21	*L. monocytogenes*	4 b	Blue cheese enstablishment (this research)

## Data Availability

All the data presented in this study are available within the article.

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
