# Peer review of "Characterization and In Vitro Efficacy against Listeria monocytogenes of a Newly Isolated Bacteriophage, ΦIZSAM-1"

_microorganisms, 2021, doi:10.3390/microorganisms9040731_

Round 1
Reviewer 1 Report
Review report
The manuscript ”Characterization and in vitro efficacy against Listeria monocytogenes of a newly isolated bacteriophage, ɸIZSAM-1” by Scattolini and colleagues introduces a new bacteriophage infecting Listeria monocytogenes. The study describes a bacteriophage isolated from a blue cheese plant and that infects several Listeria strains. in vitro efficacy test, protein profile and part of genome data are introduced. The genome sequence and annotation results are not included in the MS as they are under evaluation for patent. The paper would have benefitted remarkably from the whole genome sequence.
In general, the data has been produced with sufficient methods and the paper is mainly written in an understandable way. However, there are some parts that need revision.
Abstract:
“and showed lytic activity against anti-microbial resistant Listeria monocytogenes strains,
in particular.” Please remove in “particular”.
Methods:
The authors refer to centrifugation speeds with g and rpm – please use an consistent term.
Results and discussion:
The authors state that no phage resistant hosts were detected – however it is not clear how this conclusion has been made. 33 hours seem to be too short of an period to state this.
“The two genomes were co-linear and pairwise alignment showed a global identity of about 46%.” Please change co-linear -> linear. What was the coverage of this region that showed 46% identity? What is the genome length of the closest relative phage?
“…for a total molecular weight of 1 445.5 kDa” – this does not seem to be relevant information as proteins may occur in multiple copies in phage particles.
More specific corrections for the manuscript should include:
Throughout the text, super- and subscripts should be corrected, as well as the terms that are usually written in italics (such as in vitro, i.e., and scientific names of organisms).
The authors refer to the bacteriophage with both ɸIZSAM and bacteriophage IZSAM – please use an consistent term when referring to the phage.
Figure 2 legend: change no-treated -> untreated
Page 6: …“was verified to be more successful that the “passive mode”,… that -> than
Page 7: first line: led -> lead; significantly ->significant
Author Response
Title
Characterization and in vitro efficacy against Listeria monocytogenes of a newly isolated bacteriophage, ɸIZSAM-1
Review report
The manuscript “Characterization and in vitro efficacy against Listeria monocytogenes of a newly isolated bacteriophage, ɸIZSAM-1” by Scattolini and colleagues introduces a new bacteriophage infecting Listeria monocytogenes. The study describes a bacteriophage isolated from a blue cheese plant and that infects several Listeria strains. in vitro efficacy test, protein profile and part of genome data are introduced. The genome sequence and annotation results are not included in the MS as they are under evaluation for patent. The paper would have benefitted remarkably from the whole genome sequence. In general, the data has been produced with sufficient methods and the paper is mainly written in an understandable way. However, there are some parts that need revision.
Re: We thank the valuable reviewer for the comments to our draft paper. We made all the corrections suggested to make the paper clearer and more understandable.
The Report on Zoonosis (2021) was updated with the most recent published one (end of February) and relative to 2019 (reference n. 1).
On behalf of all the authors, I hope this editing could have improved the quality of our paper and I kindly thank the reviewer for the very precious suggestions.
Abstract:
“and showed lytic activity against anti-microbial resistant Listeria monocytogenes strains, in particular.” Please remove in “particular”.
Re: Thank you. “in particular” was removed
Methods:
The authors refer to centrifugation speeds with g and rpm – please use an consistent term.
Re: Thank you, we expressed all centrifugation speeds with g, and we changed the values according to conversion factors.
Results and discussion:
The authors state that no phage resistant hosts were detected – however it is not clear how this conclusion has been made. 33 hours seem to be too short of a period to state this.
Re: thanks to valuable reviewer for this comment. The sentence was changed in order to give explanations (as data not shown) about some preliminary results on phage resistance in relation to our experiment. However it was also stated that 33h time is quite a short period to give any exhaustive conclusions. More in-depth studies will be needed to confirm this finding.
“The two genomes were co-linear and pairwise alignment showed a global identity of about 46%.” Please change co-linear -> linear. What was the coverage of this region that showed 46% identity? What is the genome length of the closest relative phage?
Re: Thank you for the valuable comment. The word “co-linear” was changed in “linear” and the coverage was added.
“…for a total molecular weight of 1 445.5 kDa” – this does not seem to be relevant information as proteins may occur in multiple copies in phage particles.
Re: Thanks for this very important consideration. Any reference to a total molecular weight was removed from the text and the abstract edited accordingly.
More specific corrections for the manuscript should include:
Throughout the text, super- and subscripts should be corrected, as well as the terms that are usually written in italics (such as in vitro, i.e., and scientific names of organisms).
Re: Thank you for the precious remarks. By transferring our text into MDPI format, all the words needed to be re-checked more accurately. So we corrected throughout the text super-and subscripts and we formatted in italics the elements that you mentioned (in vitro, i.e. and the scientific names of microorganisms).
The authors refer to the bacteriophage with both ɸIZSAM and bacteriophage IZSAM – please use an consistent term when referring to the phage.
Re: Thank you for the precious comment. The authors preferred to use the symbol “ɸ” before phages’ names. However we tried not to start new sentences with that symbol, so the authors changed the sentences accordingly.
Figure 2 legend: change no-treated -> untreated
Re: Thank you. We changed the adjective
Page 6: …“was verified to be more successful that the “passive mode”,… that -> than
Re: Thank you. We corrected the preposition.
Page 7: first line: led -> lead; significantly ->significant
Re: Sorry for the mistakes. We corrected the past of the verb and we changed the adverb with the adjective.

Reviewer 2 Report
Listeria is an important foodborne pathogen that causes listeriosis, an infection that in some cases may have an fatal outcome. Numerous Listeria bacteriophages have been described so far, however a complete characterization is available only for a small number of the discovered bacteriophages. Moreover, data on genome sequences are scarce. The control of L. monocytogenes in the food processing setting has proven difficult due to its great ability to adapt in various environmental conditions. Recent research on bacteriophages as natural antimicrobial agents has revealed that they can be useful for targeting the bacterial pathogens in different food matrices. The study by Scattolini et al. represents a valuable work given that they have described the phage ɸIZSAM-1 for L. monocytogenes biocontrol. Moreover, they propose a new approach to phage isolation for applications in Listeria monocytogenes biocontrol in food productions
Please check the grammar (e.g. In particular, the author believe that the selection of phages from the same environments where pathogens live represent the ideal way to successfully integrate the control measures in an innovative, cost effective, safe and environmentally friendly new way). There are also some typos in the manuscript text (e.g. Figure 1. … integration form the source…).
Listeria monocytogenes should be written using the italic style. Please check this in whole manuscript.
Author Response
Title
Characterization and in vitro efficacy against Listeria monocytogenes of a newly isolated bacteriophage, ɸIZSAM-1
Review Report Form
Comments and Suggestions for Authors
Listeria is an important foodborne pathogen that causes listeriosis, an infection that in some cases may have a fatal outcome. Numerous Listeria bacteriophages have been described so far, however a complete characterization is available only for a small number of the discovered bacteriophages. Moreover, data on genome sequences are scarce. The control of L. monocytogenes in the food processing setting has proven difficult due to its great ability to adapt in various environmental conditions. Recent research on bacteriophages as natural antimicrobial agents has revealed that they can be useful for targeting the bacterial pathogens in different food matrices. The study by Scattolini et al. represents a valuable work given that they have described the phage ɸIZSAM-1 for L. monocytogenes biocontrol. Moreover, they propose a new approach to phage isolation for applications in Listeria monocytogenes biocontrol in food productions.
Re: We thank the valuable reviewer for the comments to our draft paper. We made the corrections suggested by the reviewer.
On behalf of all the authors, I hope these editing could have improved the quality of our paper and I kindly thank for the very precious suggestions.
Moerover, the Report on Zoonosis (2021) was updated with the most recent published one (end of February) and relative to 2019 (reference n. 1).
Please check the grammar (e.g. In particular, the author believe that the selection of phages from the same environments where pathogens live represent the ideal way to successfully integrate the control measures in an innovative, cost effective, safe and environmentally friendly new way). There are also some typos in the manuscript text (e.g. Figure 1. … integration form the source…).
Re: Thank you for the suggestions. By transferring our text into MDPI format, all the words needed to be re-checked more accurately. We checked the grammar and made the sentence clearer.
Listeria monocytogenes should be written using the italic style. Please check this in whole manuscript.
Re: Thank you. We checked the whole manuscript and we formatted in italic style “Listeria monocytogenes” and all the other words where this style was needed.
